# Expanding the Mind–Body–Environment Connection to Enhance the Development of Cultural Humility

**DOI:** 10.3390/ijerph192013641

**Published:** 2022-10-21

**Authors:** Isabelle Soulé, Chloé Littzen-Brown, Amber L. Vermeesch, Layla Garrigues

**Affiliations:** 1School of Nursing & Health Innovations, University of Portland, Portland, OR 97203, USA; 2Department of Family and Community Nursing, School of Nursing, University of North Carolina Greensboro, Greensboro, NC 27412, USA

**Keywords:** cultural humility, mind–body, mindsight, yoga, nature-based intervention, forest therapy

## Abstract

The unique health care needs of diverse individuals and communities are complex. To meet these needs, healthcare professionals are being called upon to alter traditional ways of thinking, perceiving, and acting in order to create more inclusive environments. Research shows that using mindsight, a process that increases both insight and empathy, can enhance an individual’s mind–body-environment connection, increase self-awareness, and promote the development of cultural humility. This paper will discuss the current perspectives on the mind/body/environment connection from a Western lens that may impact the enactment of cultural humility for healthcare providers. Two evidence-based approaches, yoga and forest therapy, are recommended as effective intervention tools in fostering mindsight and cultural humility. Blending traditional cognitive learning with techniques anchored in the physical body may hold promise in supporting the development of mindsight and cultural humility in healthcare education and practice.

## 1. Introduction

Interest in the development of cultural humility has risen in the past 25 years resulting in an extensive body of literature to support understanding, planning, implementing, evaluating, and refining care across diverse populations. Cultural humility, a paradigm introduced in the late 1990’s as an alternative to the mainstream model of cultural competence, has three primary tenets (a) a lifelong commitment to self-reflection and self-critique, (b) identifying and leveling power differentials, and (c) building mutually beneficial, high-quality partnerships [1,2]. Cultural humility is based on the simultaneous process of discernment of self and others, an awareness of the power dynamics that underlie interactions and contexts, and an ability and willingness to attune to and learn from the lived experience of diverse individuals and communities. While cultural humility includes a mindset, this paper introduces the physicality, or physical nature of developing cultural humility, and posits that techniques anchored in the body including yoga and nature-based therapies, hold promise in enhancing the development of cultural humility in healthcare education and practice.

Honoring individuals and communities as experts in their own experience requires healthcare providers to be self-aware, open-minded, and open-hearted, blending the roles of teacher and learner. Healthcare providers who practice cultural humility demonstrate intellectual, attitudinal, and behavioral flexibility to interact in a non-judgmental way with individuals who have widely different values, beliefs, worldviews, and healthcare practices than their own [3] Western healthcare providers are often educated to think of themselves as experts, which can lead to misinterpreting or dismissing teachings that offer an alternate set of values or beliefs [1] Moreover, building partnerships where health professionals value the expertise of the client and community in their own decisions often runs contrary to how professionalism is taught and role modeled in healthcare education and practice today [1,4].

The unique health care needs of diverse individuals and communities are complex. To meet these needs, healthcare professionals are being called upon to alter traditional ways of thinking, perceiving, and acting in order to create more inclusive healthcare interactions and environments. Incorporating integrated health and nature-based strategies can support diverse groups in meeting their health and wellness goals, while also promoting the development of cultural humility in healthcare providers and systems. The following paper will discuss the current perspectives on health from a Western lens that may impact the development of cultural humility and will conclude by recommending evidence-based approaches to developing cultural humility by using innovative integrative health interventions.

### Background

Much of the research and literature promoting care of diverse individuals in the healthcare environment has emerged from the United States, and has been conceptualized, carried out, and evaluated predominantly by Western-educated, biomedical practitioners. These influential cultures share a focus on individualism, a belief system based on the separation and autonomy of individuals [1]. An individualistic perspective may not recognize how individualism undermines the social fabric of a community, the interdependence of individuals with each other, and the natural environment in which they live and interact [1]. In contrast, individuals and populations that come from a collectivist perspective perceive themselves as intrinsically part of a group and emphasize interdependence over independence, affiliation over confrontation, and cooperation over competition Collectivism recognizes the group, and not the individual, as the basic unit of survival. Both individualism and collectivism have equal, albeit different, merits. However, of pivotal importance is the understanding that each of these perspectives relies on different mechanisms and values in decision-making [1]. These differences manifest themselves in matters such as gender roles, positions of authority, sense of self and personal space, communication, relationship to time, relationship to others, learning styles, and spiritual practice. Within the healthcare system, these differences manifest in how health and illness are perceived and manifested; what is thought of as cause; how, when and where communication occurs; the roles of the health professional, client, family, and community; and how treatment is negotiated, implemented, and evaluated. Because cultural humility relies on the accurate, simultaneous discernment of self and others, healthcare providers must learn to hold individualism alongside collectivism with an ability to skillfully move between these two distinct standpoints in order to better support the needs of diverse individuals and communities.

The separation of the mind from the body in modern Western thought, in favor of the mind over affective and perceptive knowing, is woven throughout biomedicine and healthcare education [5,6]. Rooted in the philosophy of the mind, the separation of the mind and body is commonly referred to as Cartesian dualism as originally proposed by Rene Descartes [7]. This perspective perceives the body largely as a transporter of the mind, and the methods of teaching, learning, and healthcare provision as fundamentally a disembodied process [8]. When a primary effort of healthcare education and professional practice is cognitive knowing, the physical body, with its sensations, emotions, affective processes, and visceral responses, is excluded as another important way of knowing [9,10]. Physical bodies, however, are central elements in communicating and interpreting cultural norms through movements, gestures, posture, voice, scent, sound, visual cues, and proximity to one another [11]. Moreover, relational aspects of care such as compassion and empathy are far more related to affect than to cognition.

Bennett and Castiglioni (2004) [5] argue that the exclusion of the physical body leads to cognitive-based observation of an experience rather than to the embodied experience itself. These cognitive-based observations are limited in that they are separate from bodily perceptions, feelings, emotions, or personal and interpersonal interactions. When a primary focus of healthcare provision is cognitive-based observation, including analytical processing and reasoning, it limits the ability to accurately perceive the whole and interconnecting parts [5]. This limited viewpoint affects a healthcare providers ability to perceive themselves accurately, discern power dynamics present; therefore, preventing them from leveling those power differentials. Furthermore, without an embodied, physical awareness of themselves and of others, providers will be limited in their ability to build mutually beneficial, high-quality partnerships, another key tenet in the development of cultural humility.

Many scholars have argued that to successfully engage with others, knowledge alone is insufficient; one must develop a sensitivity for appropriate responses and the feeling of appropriateness (multisensory awareness) which allows for accurate discernment of the context, emotions, and meaning-making of interactions that accompanies cognitive knowledge [1]. This ability to physically experience one’s own and another’s experience is termed embodiment, including the integration of the cognitive, relational, emotional, aesthetic and spiritual aspects of human experience, and revealed through empathy for, and connectivity with diverse others. The extent of one’s ability to be aware of another person’s state of mind and emotions depends on one’s own self-awareness, and research has demonstrated that people who are more aware of their physical bodies are more empathic [10]. In addition, when individuals can sense their own internal state, the pathways for resonating with others are also open [10]. Bennett and Castiglioni (2004) [5] addressed “embodied ethnocentrism,” meaning things feel familiar and comfortable in one’s own culture. This experience can be considered the physical manifestation of ethnocentrism—the perception that one’s culture is central to reality. When steeped in embodied ethnocentrism there is a feeling of discomfort when in the presence of unfamiliar others, situations, or surroundings. Furthermore, without a similar sense or feeling for another’s context, there are limitations in one’s depth of understanding, and ability to adapt, demonstrate empathy, and ultimately build high-quality interpersonal relationships.

Recent developments in interpersonal neurobiology reveal the interconnection and interdependence of mind and body. Specifically, mindful awareness, a state that includes awareness of the inner workings of the mind, sensory systems, and visceral responses, can help regulate emotional responses by promoting empathy and openness to diverse values, beliefs, and worldviews [11]. This flexibility allows an individual to lean in to or approach an unfamiliar or challenging circumstance rather than withdrawing from it [11].

In Philosophy in the Flesh, Lakoff and Johnson (1999) [1] addressed the physical aspects of interrelatedness in this way:

There exists no… person… for whom thought has been extruded from the body. That is, there is no real person whose embodiment plays no role in meaning, whose meaning is purely objective and defined by the external world, and whose language can fit the external world with no significant role played by mind, brain, or body. Because our conceptual systems grow out of our bodies, meaning is grounded in and through our bodies (p. 6).

Given the innate link between mind and body, what is interpreted in the mind is also felt in the physical body. Therefore, fear, anxiety and uncertainly are consistently accompanied by physical tension, narrowed thinking, and skewed perceptions. These constrictions, when chronically held in the body, can limit one’s ability to perceive oneself, the world, and others accurately [1]. A recent example of substantial importance is the COVID-19 pandemic; and in particular, the experience of healthcare providers who have worked throughout the pandemic. With an unprecedented degree of uncertainty and continuously changing environments, healthcare providers have reported a significant increase in chronic anxiety, fear, and moral distress which compromises their ability to build high quality partnerships [12,13].

Developing the skills to work effectively across diverse populations requires physical and multisensory awareness, and adaptation to intellectual, attitudinal, and behavioral differences. The bodily experience of feeling comfortable across diverse cultures includes an embodied awareness that is mediated through bodily sensory experiences involving the skin, muscular, tactile, visual, auditory, olfactory, and gustatory pathways [14]. It also involves the energetic field around us that brings a sense of resonance or dissonance when in the physical proximity of diverse others. This sensibility also extends to our relationship to the built environment and sensing our connectivity to the natural world around us. [5,15]. Working in parallel with traditional cognitive learning, techniques anchored in the physical body may hold promise in adding support to the development of cultural humility in healthcare education and practice.

## 2. Embodied Understanding: Mindsight

A new field of study, interpersonal neurobiology, integrates a variety of sciences with alternate ways of knowing that has helped illuminate the subjective world of the mind. Based on research investigations of affective neuroscience, an illustrative process has been developed to enhance an individual’s ability to open a more adaptive and flexible way of being [11]. Daniel Siegel’s (2010) [10] theory of “mindsight” is a process that has been found to increase both insight and empathy and is at the foundation of both social and emotional intelligence. Mindsight, based on ancient wisdom traditions of mindfulness and contemporary interdisciplinary discovery and brain research, can be cultivated through practical steps, can change the physical structure of the brain, and activates the circuits that create resilience and well-being, as well as those that underlie empathy and compassion [10]. Mindsight is best thought of as a kind of focused attention that allows observation of the internal workings of one’s own mind and helps to perceive mental processes that may be typically outside conscious awareness. This practiced observation helps us to understand ourselves as unique cultural beings, including our distinctive blend of beliefs, values, and worldviews, and the larger context in which they have developed [1]. Mindsight shifts us from automatic processing to illuminate ingrained behaviors that may not be well suited to building rapport with people of diverse backgrounds. The ability to understand one’s own mind as a parallel process to accurately discerning the inner world and contexts of others is essential in building rapport and working effectively across diverse worldviews. If one is unable or unwilling to make these shifts, then entrenchment, judgment, dividedness, and exclusion ensue. Becoming conscious of one’s patterns of the mind, that are enacted through the physical body, stimulates intrapersonal growth which also facilitates interpersonal growth in an iterative cycle [1].

Neuroplasticity is the capacity for creating and growing new neurons and neural connections in response to experiences [16]. This lifelong process is relevant in the development of cultural humility because it is foundationally about being able to understand one’s own reality and having genuine empathy for the experience of another, albeit significantly different from one’s own. The physical sense of empathy includes not only an intellectual connection but an attunement through facial expressions, tones of voice, gestures, and postures—some so imperceptible that they can only be seen on a slowed-down recording. It can be a palpable sense of connection and aliveness; for example, one client referred to it as “feeling felt,” [10], (p. 10). This skill to monitor and modify one’s internal world is termed mindsight [10] and can be conceived as the embodied and relational process that leads a healthcare provider toward self-awareness, a sensibility of the power dynamics at play in any given context or interaction, and an ability to build mutually beneficial partnerships with diverse patients, families, and communities. Embodied practices such as mindsight can be a powerful way to integrate consciousness and offers hope for evidence-based processes that promote open-mindedness, open-heartedness, attunement, and a felt sense of empathy, understanding and connectivity with others [10].

### 2.1. Yoga as a Tool to Develop Mindsight and Cultural Humility

An example of a mindful practice that shows promise in enhancing the development of mindsight and cultural humility, is yoga. Yoga, an ancient wisdom tradition, can be traced back to the third century BCE and is rooted in Indian philosophy [17,18]. In traditional yogic texts, yoga is defined as the union of the individual with the Absolute (e.g., a higher power); a pathway that allows for such a union, and an unruffled state of mind under all conditions [19]. In contrast, in western society, yoga is most often thought of as a physical practice focused on yoga postures (asana), with or without breathing exercises (pranayama), or meditation (dhyana). Yoga practice from either of these perspectives slows down physical movement, brings focused attention to the physical body, and its corresponding feelings and sensations, raising conscious awareness of the mind–body–environment interaction. Learning to be fully present in the here and now can be used to support the connection between mind/body/environment, therefore enhancing the development of self-awareness, an essential component in the lifelong development of cultural humility.

As a mind–body intervention, yoga has shown promising results for a variety of human concerns, including: healthy eating habits [20]; asthma [21]; cancer symptom management [22]; cardiovascular disease [23]; menopause [24]; multiple sclerosis [25]; pain [26,27]; and mental health conditions [17] such as anxiety [28], depression [29], and post-traumatic stress disorders [30]. Scientists have postulated how participating in yoga as an intervention can have such positive impacts across the human condition, albeit with limitations. By means of polyvagal theory [31], yoga has been claimed to facilitate eudemonic well-being and physical, mental, and behavioral health benefits for diverse populations through enhanced self-regulation and resilience cultivation [32]. Similarly, yoga has been proposed to optimize the human hypothalamic-pituitary-adrenocortical axis by regulating cortisol levels, increasing blood flow in the brain, and promoting neurogenesis and synaptogenesis, ultimately increasing neuroplasticity [33].

Yoga is posited as a tool to help promote mindsight and cultural humility, and is of particular importance at this time in history, as healthcare providers have reported suboptimal well-being at work due to the numerous implications from the COVID-19 pandemic. Developing a mind–body–environment connection is increasingly important to protect and enhance well-being, especially during times of separation, stress, and sickness, hallmarks of working through the COVID-19 pandemic. Recent research supports that suboptimal well-being at work is a significant correlate for several negative organizational (e.g., patient medication errors) and personal outcomes (e.g., well-being) [13,34]. These suboptimal states of duress can limit nurse’s ability to develop self-awareness, which then limits their ability to effectively build high-quality partnerships across differences. While more research is needed, yoga shows promise in promoting the mind–body–environment consciousness necessary for developing mindsight and cultural humility and improving both individual well-being and interpersonal relationships.

### 2.2. Forest Therapy as a Tool to Develop Mindsight and Cultural Humility

Another example of a mindful practice that can be used in developing mindsight and cultural humility, is the nature-based intervention of forest therapy, also referred to as forest bathing or shinrin-yoku. Forest therapy is a way to connect oneself with the natural world in order to increase the mind–body-environment connection [11]. There are various definitions for nature depending upon the context and culture of individuals and populations; however, for this paper, nature is defined as an environment, either urban or rural, that has naturally grown plants or plants grown by humans [35]. Forest therapy includes both active and/or passive participation including being in the presence of and viewing nature or direct, active engagement with nature. According to the Association of Nature and Forest Therapy, forest therapy is a “relational practice that brings people into deeper intimacy with natural places,” [36], (para. 1). Additional considerations for forest therapy include the idea that one’s own physical body is a natural place [37]. Therefore, engaging in forest therapy aids in strengthening the mind–body–environment connection. Strong scientific and objective evidence exists for ways in which forest therapy supports multiple aspects of well-being, including reduced stress levels, increased immune response and function, improved cardiovascular and respiratory health, attention restoration and a reduction in depression symptoms [38,39].

Forest therapy has been found to produce positive benefits for individuals engaging in the natural environment including positive health, increased energy, increased sense of purpose, and a decreased sense of stress [40,41,42,43,44]. In addition, there have been numerous studies investigating stress reduction through cognitive behavioral therapy and mindfulness applied in a forest environment. For example, response rate for stress reduction interventions includes baseline cognitive behavioral therapy and mindfulness in natural settings outdoors or indoors with plants with stress reduction in six weeks [45]. Kim et al. (2009) [39] investigated the application of a four-week forest-walking based cognitive-behavioral therapy program for treating clinical depression. The results showed a significant decrease in depression after the nature intervention. Morita et al. (2011) [46] investigated two-hour forest-walking sessions conducted on eight different weekend days, which provided reductions in feelings of depression, anxiety, and stress. Immersive experiences in nature have been found to improve mindfulness, rejuvenate one’s body, mind and energy, and reduce physical, emotional, and cognitive stress [44] while deepening respect toward the natural environment. These improvements allow for productive changes in attitudinal, intellectual, behavioral flexibility, and bring internal clarity and calmness, which are foundational tools for successful engagement with self, others, and the natural environment [47].

Two theoretical frameworks, stress reduction theory and attention restoration theory further demonstrate the connection between forest therapy and the development of mindsight and cultural humility. Stress reduction theory, originally proposed by Ulrich [48] postulates that in the presence of nature, there is a negative association with stress, meaning that when in a natural environment, perceived stress decreases. Furthermore, as individuals experience a reduction in stress through nature exposure (i.e., forest therapy), their capacity for mindsight increases, which in turn opens the channels for self-awareness and the development of cultural humility. Similarly, attention restoration theory as originally proposed by Kaplan and Kaplan postulates that with nature exposure there is a positive association with attention, meaning that in the presence of nature, attention increases [35]. It can then be postulated that with a reduction in stress and an increase in attention, the context is fitting for the development of mindsight and cultural humility.

## 3. Conclusions

The development of mindsight and cultural humility are a vital set of skills necessary for healthcare providers in contemporary professional practice, requiring a lifelong commitment to deepening self-awareness, leveling power differentials, and building mutually beneficial high-quality partnerships with diverse others. Current Western perceptions of individualism in addition to the COVID-19 pandemic have generated a mind–body disconnection for healthcare providers, and interventions such as yoga and/or forest therapy show promise in helping to reconnect with oneself, develop high-quality interpersonal relationships, as well as a deeper connection with the natural world.

Because the mind and the body work in synchrony to receive and interpret sensory information, blending cognitive and embodied abilities can enable perception with increased depth, clarity, and accuracy. Yoga and forest therapy are recommended as established ways to improve interconnection between body, mind, and the natural environment in addition to reducing stress and anxiety, increasing focused attention, and improving self-awareness. Noting the physical nature of the development of cultural humility, further research is needed to understand the context in which self-awareness is best developed; strategies to promote self-awareness; examination of the varying capacity, propensity, and/or desire of individuals to develop self-awareness, and reliable and valid measures for studying the application of self-awareness on the development of cultural humility as it is manifested in an educational setting, clinical encounter, or in a healthcare system.

## Data Availability

Not applicable.

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
