# Peer review of "Expanding the Mind–Body–Environment Connection to Enhance the Development of Cultural Humility"

_ijerph, 2022, doi:10.3390/ijerph192013641_

Round 1
Reviewer 1 Report
I have appreciated reading the article about cultural humility, connection between body and mind, and yoga and forest therapy as possible boosts of cultural humility.
While the article presents interesting information, I have major concerns about the current version of the manuscript.
The article presents no empirical research conducted by the authors and is not based on a systematic review of literature. Also, there is not a true theoretical integration of previous literature, but it rather sounds that the authors have decided to select information supporting their arguments.
Language often sounds not scientific, especially on the section about connection between body and mind. I would encourage authors to summarize as precisely as possible literature showing such interconnection and to pay attention to neutral, scientifical language.
The authors have summarized advantages of yoga and of forest therapy but have not clarified why such benefits could extend to cultural humility. If the authors claim that yoga and forest therapy could be useful tools to boost cultural humility, they should explain in detail why (e.g., via which processes? which cultural humility components would be boosted and why?) yoga and forest therapy could be useful for cultural humility.
Also, detailed and deep explanations about why yoga and forest therapy can improve interconnection between body and mind are missing.
While the article presents no empirical research or systematic literature review, there are no future research propositions.
The sections of the article are not well connected. For example, cultural humility is explained in the beginning of the article, but then almost not mentioned until the conclusion. Then why would information in the middle of the article be related to cultural humility?
Minor points:
It is not clear whether all the MBCT programs mentioned are based on forest therapy. This should be specified. Otherwise, what do they have to do with the paragraph on forest therapy?
There are minor punctuation imprecisions (e.g., points missing).
I hope these comments will be useful to the authors.
Author Response
Please see attached for our response to reviewer 1

Reviewer 2 Report
I reviewed a short article that essentially has three parts: 1) introduction and background, establishing the importance of the topic: why healthcare providers (in this case, nurses) can find their lives and their job outcomes greatly enhanced with the development of "mindsight". This part briefly but expertly leads the reader through the philosophical basis of a Western bias towards "mind over matter" and isolating individualism to a consideration of how this cultural modality can result in a spiral of stress, depression, and closing off. Hence, an inability of a caretaker to "feel" or interact with humility/empathy to the self and others. The twin notions of cultural humility and mindsight are covered as existing modes of reestablishing the possibility of effective interpersonal communication and care. Mindsight is the opposite of "embodied ethnocentrism," therefore we might call it embodied cultural competence/cross-cultural empathy. The second part (2) briefly describes the potential of yoga to train mind-body awareness as a foundation for mindsight; and part (3) the potential of "forest therapy" (mindful being in natural environment) as another therapy with some demonstrated positive outcomes.
Overall: This article is exceptionally well-written. The introductory pages provide a very clear explanation for how a culture of healthcare that privileges the provider's "authority" and "rationality" can lead not only to subpar patient experience, but also cause almost unbearable "burnout" and failure to connect (with patients) among providers. The scientific evidence for the effectiveness of cultivating balancing modalities in providers (mind-body awareness, embodied capacity for empathy, etc.) is presented with equal clarity.
An important contribution to existing literature, delivered in clear and organized prose, without jargon. Short, compelling, accurate presentation.
I have no suggestions for improvement. This paper is a mature product of careful and well-documented methodology. Where necessary, the authors summarize the relevance of experiments conducted (with citations) to show cognitive benefits of yoga and/or forest therapy.
The conclusions are entirely consistent with the main argument presented. The line of reasoning is very clear.
I do not see any reason that this paper – which is articulate, well-organized, and convincing in its argument – should be revised.
Footnotes cite relevant and scientifically legitimate sources.
I would reiterate that this paper is flawless in terms of English language prose.
Author Response
Please see attached for our response to reviewer 2

Reviewer 3 Report
Dear Authors,
Thank you for the opportunity to read your manuscript dedicated to such an interesting and extremely important topic in the current context.
The manuscript is well written and organized according to the criteria of a scientific article. However, there are some gaps that it would be important to fill. Namely, the absence of an empirical study or, at least, of a project to implement a study with this component. This absence is also felt in the conclusion, where no mention is made to the continuity of the study of this topic or the application of these techniques (as Yoga and Forest therapy) in real contexts. This lacks should be mentioned in the final version of the paper.
In methodological terms, the article is based essentially on a literature review. As such, it would be useful to present a summary table of the results obtained in previous research that focuses on the area under study.
Author Response
Please see attached for our response to reviewer 3

Reviewer 4 Report
This is an incredible paper. Not that I would necessarily add to it, but while reading it I thought about how much the integrative perspectives discussed fit in with the United Nations Sustainable Development Goals that link good health and well-being (SDG 3), with respecting cultures (SDGs 11.4 and 16), and preserving our forests so we can have forest therapy (SDG 15).
Very inspirational article.
Author Response
Please see attached for our response to reviewer 4

Round 2
Reviewer 1 Report
I have read the revised version of the manuscript about yoga and forest therapy as possible tools to foster body-mind-environment interconnection and cultural humility.
The authors have revised the conclusions of the article, adding some explanations about why yoga and forest therapy could contribute to development of cultural humility, and suggesting future research to investigate such associations. However, the paragraph on future research suggestions should better be expanded and be more specific and concrete.
I see why a systematic review of literature is not possible – as the authors explained in the cover letter – but this should be explicitly and clearly stated in the manuscript.
Besides answers to these requests, my other suggestions were almost ignored, and this version of the manuscript (except the conclusions) is almost identical to the previous one.
The sentences that – according to the cover letter – should answer to my call for explaining why yoga and cultural humility could promote mind-body interconnection and cultural humility were already there in the previous version.
I asked the authors to clarify whether the mentioned MCBT programs are based on forest therapy (lines 370-377 in the revised version) but no modification was made.
There was no effort to improve connections between sections, besides just mentioning cultural humility in lines 249 and 347.
Author Response
Please see attached document for authors' response to review 1.
